

# Culture matters: Factors influencing natural hazard risk preparedness – a survey of Swiss households

Elisabeth Maidl[1,2], David. N. Bresch[3,4], Matthias Buchecker[1]

[1]Swiss Federal Institute for Forest, Snow and Landscape Research WSL, Birmensdorf, Switzerland
[2]University of Applied Sciences and Arts Northwestern Switzerland, Olten, Switzerland
[3]ETH Zurich, Department of Environmental Systems Science, Zurich, Switzerland
[4]Federal Office of Meteorology and Climatology MeteoSwiss, Zurich, Switzerland

*Correspondence to*: Elisabeth Maidl (elisabeth.maidl@wsl.ch)

**Abstract.** Building a culture of risk is an essential objective within the integrated risk management paradigm. Challenges arise both due to increasing damage from natural hazards and the complexity in interaction of different actors in risk management. In Switzerland, the Strategy for Natural Hazards Switzerland, aims to establish efficient protection of the population, natural resources and material goods. This requires that all actors are recognized and aware of their responsible role in risk management. However, previous, non-representative studies indicate that risk awareness and preparedness levels are rather low within the general population. For the first time, our nation-wide survey provides empirical data on factors that influence individual risk preparedness. Multivariate analysis shows that taking responsibility for natural hazard risk prevention is not only related to personal experience and perceived probability of hazard events, but also crucially influenced by social forms of communication and integration. Therefore, we conclude that social capacity building needs to include such factors in order to render integrated risk management strategies successful.

## 1 Introduction

Natural hazard risk mitigation is of increasing importance in Alpine states and ranks high on the political agenda. Meeting the challenge of rising damage levels requires overcoming the paradigm of hazard-based risk management in favour of integrated risk management (IRM) strategies. IRM aims for establishing a culture to live with hazards. It is a comprehensive normative concept embracing the following elements: (1) consider all kinds of natural hazards, (2) monitor and evaluate risks, including the effect of prevention and preparedness, and (3) involve all relevant players, i.e. decision-makers and stakeholders (PLANAT, 2004). To promote these objectives, social capacity building is at the core of both, practice and research on natural hazard risk management. The mobilization of social resources in risk management is increasingly recognized in research (e.g. CapHaz-Net, KultuRisk), and centrally emphasized in strategic documents of international organisations (e.g. Sendai Framework for Disaster Risk Reduction), as well as Swiss agencies (Aller and Egli, 2009; PLANAT, 2004). So far, the understanding how to activate such resources is limited. This paper therefore addresses the research question how to raise risk preparedness in the general population. Management measures need to connect to the population's available capacities to



perform individual risk preparedness (IRP). This study provides evidence on influencing factors on IRP using the concept of social capacities (Aven and Renn, 2010; Kuhlicke et al., 2011).

The study design is based on methods and results of a previous local survey (Maidl and Buchecker, 2015) adjusted for usage on a nation-wide level, which covers a diversity of cultural and geographical regions as reflected by four official languages, and comprehensive types of natural hazards. As far as methodology is concerned, we focus on transparency and multidimensionality of measurement methods, combining key concepts and question wording as used in earlier studies (e.g. Bubeck et al. 2012; Thieken, 2007). In addition to the analytical and methodological purpose, the study aims to provide relevant insights for practitioners. Practitioners in IRM need to understand how they can best motivate private actors to engage in protective behaviour. It remains a key issue that for average citizens, natural hazards play a minor role in their daily lives. In research on environmental behaviour (Steg and Vlek, 2009) the dilemma of public goods at risk in a situation of low personal benefit is well known. Following this logic, personal relevance of natural hazards would be a key pre-requisite for taking preventive action.

This study for the first time provides nation-wide data to answer this and more questions regarding a broad scope of potential influence factors on IRM like risk awareness, experience, information or personal values.

## 2 Risk preparedness: underlying concepts and theory

### 2.1 Integrated risk management

International organizations and national agencies increasingly emphasize social capacity building as a core element of disaster risk reduction (DDR). Alpine regions stand out in developing risk management strategies that increasingly take into account social aspects of risk management (for Switzerland: PLANAT, 2004; FOCP, 2014; for Austria: BMLFUW IV/5, 2012). This indicates a shift away from the focus of traditional risk management on technical hazard control that mostly relies on structural mitigation measures, towards integrated risk management (IRM). The focus lies on non-structural measures like land use planning, legal frameworks, organizational measures, and risk communication (Merz et al., 2010). A previous study in Switzerland showed that the population favours non-structural measures with similar priority as traditional hazard mitigation (Buchecker et al., 20131, 2016). However, there is no empirical evidence in Switzerland of how well these attempts are developed, and what capacities actually are required to maintain and strengthen them. Research on capacity building in European countries is scarce, and in the case of Switzerland, only small-scale studies within certain population segments are available (Siegrist and Gutscher, 2006; 2008; Buchecker and Maidl, 2015).

In Switzerland, political, professional and private actors share responsibilities in natural hazard risk management (Hess, 2016). On a federal level, legislated common guidelines such as the National Strategy for Risks Management (PLANAT, 2004) are developed which support research, education, and early warning. The cantons are responsible for implementing laws. Local land use and emergency planning takes place at municipal level. On the side of private actors, property owners are responsible for taking protective measures. In most cantons, building insurance is mandatory, and, according to the hazard map, preventive



measures are required. A natural hazard map is available in each municipality nation-wide (Bründl, 2009). The map shows on a scale of five risk zones whether an object is located in a no-risk (white), residual (yellow-white), low (yellow), increased (blue), or high (red) risk zone. Research so far indicates that the hazard risk maps are not well known among the population (Siegrist and Gutscher, 2006; Maidl and Buchecker, 2015). The primary goal of communication strategies is therefore to raise

awareness and enhance dialogue between relevant actors (Hess, 2016).

## 2.2 Social capacity building

For the purpose of this study, Kuhlicke et al.'s (2011) definition of social capacity is applicable as it highlights the individual level as specific component: "By social capacity we mean all the resources available at various levels (e.g. individuals, organisations, communities) that can be used to anticipate, respond to, cope with, recover from and adapt to external stressors

(e.g. a hazardous event)." Therefore, capacity building means to develop all such possible kinds of resources. In their overview, Kuhlicke et al. describe six types of social capacities: motivational, knowledge, networks, institutional, economical, and procedural capacities. We particularly focus on risk preparedness as a motivational capacity, while other capacities are investigated as influencing factors. The set of these possible influencing factors is refined in reference to Höppner et al. (2012; Buchecker et al., 2013b), who additionally point to the aspect of risk acceptance, which we included in our survey. Further, we

consider social integration and social capital variables such as trust and social integration (Putnam, 2000, 2001) as concepts related to capacity building.

Turning to strategies how build social capacity, the most common intervention form is risk communication as a means to 'inform, persuade and facilitate public support for hazard risk mitigation and preparedness' (Sanquini, 2016). The basic logic of risk communication is that a change in knowledge by providing information would motivate the target population to change

their behaviour. The assumption of a direct link between information and behaviour, however is challenged in the present study. The assumption of such link roots in the deficit model of risk communication (Demeritt, 2014), according to which the target population needs to be educated to compensate its deficiency. However, information receivers proof to be rather reluctant to adopt messages from one-way communication (Maidl and Buchecker, 2015). We assume that this is not due to a lack of rationality, but rather indicates that risk-related behaviour is driven by other goals/rationales and also influenced by other factors

than information.

## 2.3 Risk preparedness

The term 'risk preparedness', according to the United Nations International Strategy for Disaster Risk Reduction (UNISDR) is defined as "The knowledge and capacities developed by governments, professional response and recovery organizations, communities and individuals to effectively anticipate, respond to, and recover from, the impacts of likely, imminent or current

hazard events or conditions." (UNISDR, 2009). This defines preparedness as a combination of social capacities and reads similar to the above introduced IRM principle and the definition of social capacity. Many of these concepts and definitions used in risk research lack distinct meanings (Shreve and Fordham, 2016), which reflects the change of paradigms and rationales



over time. During the 1970s, the term strongly related to structural measures. Then Paul Slovic laid the ground for a psychometric approach bringing in the factor of individual perception (Fischoff et al., 1984; Slovic, 1987). Emerging from criticism of the psychological approach, new conceptual frameworks aim to recognize the complexity of social, environmental, and cultural processes that may influence people's risk-related perception and behaviour (Douglas and Wildavsky's, 1982;

Dombrowsky, 1998). Social constructionism has found its way into risk research and developed further, especially in the last decade (Fichter et al., 2004; Powell and Colin, 2009; Wachinger and Renn, 2010). As a prominent instance, the interdisciplinary social amplification of risk framework (SARF) was developed as an approach (Kasperson et al., 1988; Breakwell 2007; Renn, 2008), which considers a combination of physical consequences interacting with psychological, social, institutional, and cultural processes to investigate preparedness. Earlier risk perception research had equated cognitive judgments with emotional

responses, however, the relationship between judgments and emotional reactions needs to be investigated (Wilkinson, 2001). Protection Motivation Theory (Rogers 1975; 1997) brings in perceived efficacy of protection measures in addition to the perception of threat. We assume that prior to such cognitive judgements, there needs to be a sense of responsibility. Perceived self-responsibility and privatization of risks (Steinführer et al., 2008) may be as a missing link in the explanation of behaviour. We assume that taking responsibility is more likely, if people are treated as responsible actors whose needs and concerns are

taken serious. However, risk dialogue so far is rather a dialogue among experts, and therefore calls for more open a discourse (Pearce 2003; Geoffrey et al., 2016).

## 2.4 Social capital: trust and integration

Putnam's (2000, 2001) understanding of social capital refers to features of social organization, such as trust, norm, and networks that improve the efficiency of society by facilitating coordinated actions. In the context of this study, it is assumed that a high

level of civic engagement would correspond to a high level of self-responsible and preventive behaviour. This assumption is further inspired by Chiu et al. (2013), who claimed that individual networks influence sharing of knowledge and attitudes. Trust in Putnam's sense of generalized trust is further understood as a societal resource. This source comprises two components: social trust and confidence (Sütterling and Siegrist, 2014). Trust comes into play, when confidence is no longer given, e.g. if one is confronted with the limits of hazard control. In the present context, we speak on trust only. It is operationalized as the

respondents' belief in controllability by public measures and protection from damage by authorities by multiple item scales.

To sum up the section on theoretical concepts and current gaps in research, this paper examines influencing factors on IRM as a social capacity in the context of an Alpine state. It addresses the following research questions:

(1) What factors influence individual natural hazard risk preparedness?
(2) How can risk preparedness be measured multi-dimensionally?
(3) How can capacity building practitioners make use of these insights?



## 3 Methods

### 3.1 Survey description

Questionnaire design was mainly based on our previous study on flood risk awareness and preparedness among homeowners in the city of Zurich (Maidl and Buchecker, 2015), and adjusted to the practice of hazard risk management in the whole of

Switzerland. The questionnaire was conducted in the three national languages German, French and Italian, and refers to all types of natural hazards that occur in Switzerland. We integrated the view of Swiss risk management practitioners from several cantons and federal agencies into the questionnaire design during a participatory workshop conducted in September 2014. After an iterative process of questionnaire design, a pre-test was run using a random sub-sample of our household sample (n=100, response rate= 13%). The final survey was conducted in two rounds between February and June 2015 and administered by post

mail to a random sample of the Swiss population including persons with a unlimited residence permit (N=10'000). The representative random household sample was provided by the Swiss Federal Statistical Office (FSO). Most addressees in the sample (N=8'948) are located in areas with no significant hazard risk, and 1'599 live in a risk zone, which mirrors the distribution of the general population. The response rate was 20% (n=2'137), and evenly distributed among the risk zones, which are not well known among the general population. One third of the respondents did not know which risk zone applied to

their area of residence whilst others over- or underestimated the risk. However, the response rate was higher in mountainous regions, and highest among property owners, who are over-represented with a share of 52% of respondents compared to the ratio of residential property of 37.4% in the general population (FSO, 2010). Respondent's age ranges between 18 and 85, with an average age of 52, 24% higher than the average age of the Swiss population (42 years according to the FSO). Further, highly educated respondents are over-represented compared to the general population, which is typical for paper-based mail surveys.

Female and male respondents are equally distributed.

The questionnaire comprised altogether 182 items covering 22 concepts (Appendix A1X). Each concept was multi-dimensionally operationalized using several items. Principal component analysis (PCA) was conducted to reduce the complexity of the data and to construct summated indices for multivariate analysis. Then a linear regression model with IRM as dependent variable was applied.

### 3.2 Operationalization of key concepts

### 3.2.1 Risk preparedness

Literature review showed that natural hazard risk preparedness is measured in many ways. In some studies, a one-dimensional measure was applied, mostly operationalized as the intention to adopt a particular behaviour, e.g. invest in temporary protection equipment like sandbags or take out insurance (Botzen et al.,, 2009a; 2009b). Others operationalize risk preparedness as already

adopted behaviour (Lindell and Hwang, 2008; Miceli et al., 2008; Grothmann and Reusswig, 2006). Partly due to diverging measurement methods, no validated set of influencing factors could be identified so far. According to Shreve at al. (2014), the diversity of results mirrors the diversity of circumstances of risk preparedness. They emphasize the dynamic character of



influences on natural hazard risk-related behaviour and doubt that stable influences can be identified that sufficiently account for perceptions, attitudes and vulnerabilities. This discourse in mind, operationalization of risk preparedness in this study comprises 18 items (see Appendix A1X) that represented intention to prepare for an event, as well as behaviour that was already adopted by the respondents.

### 3.2.2 Risk awareness

As in the case of risk preparedness, we found different ways to measure risk awareness. Often, it is equivalently used to risk perception, for instance as 'people's judgements and evaluations of hazards they (…) are or might be exposed to' (Rohrmann, 2000). In our view, such judgements and evaluations are most notably about the perception of probabilities. For the average citizen it is challenging to translate the scientific concept of probability into relevant meaning. Further, risk awareness is often measured as perceived danger to suffer personal damage.

Our questionnaire comprises not only these two dimensions, but altogether 27 items address risk awareness. PCA was conducted for scale construction (Appendix Table A_II). This revealed three distinct components: (1) relevance of natural hazards (including concern), (2) perceived probabilities of different hazard types in the respondents' region, and (3) perceived threat.

### 3.2.3 Natural hazard experience

Personal natural hazard experience was identified as an explanatory variable for awareness and preparedness in previous studies (e.g. Bubeck et al., 2012; Mishra and Mazumdar, 2015). It is assumed that the experience of suffering damage increases the readiness to protect oneself (e.g. Weinstein et al., 2000). However, Terpstra (2011) found that it is necessary to differentiate between qualities of experience, i.e., the kind of emotions associated with it. These can be negative emotions such as a feeling of powerlessness and result in resignation, but also positive emotions, like fascination, or to be able to prevent more severe damage. We measured different types of experiences ranging from knowing natural hazards from the media only, experience as volunteer/professional, personal endangerment, or material damage. PCA revealed two dimensions of experience: personal danger and material loss. The quality of experience was measured using separate items like self-reported effect on awareness and preparedness (Table A3).

### 3.2.4 Social capital: trust, social integration, and responsibility

Previous research showed that trust in public risk management (Terpstra, 2011; Visschers et al., 2008) and social integration (Akama et al., 2014) influence individual risk preparedness. In reference to Putnam (2000, 2001, later: Lochner et al. 2003), trust and integration are regarded components of social capital. We focused on trust in public hazard protection and measured it using a 5-item scale was chosen (Appendix Table A4).

Additionally to social capital, we investigated social integration (Appendix Table A5). Social integration in this sense touches civic engagement and is understood as a part of political culture.



The perception of responsibility was measured related to private and public actors (Appendix Table A5). Respondents also rated to what degree these actors fulfil their responsibilities.

### 3.2.5 Risk communication

The set of questions on risk communication comprised 20 different means of getting informed about natural hazards (Appendix Fig. A1). Besides traditional weather forecasts and the use of mass media, additional communication means were considered, e.g. one-way communication (information campaigns, notifications from the authorities) dialogic communication (experts, insurances, private persons; participation in trainings), visiting websites, social media, as well as usage of printed material like books, leaflets or visiting exhibitions.

## 4 Results

In section 4.1, we report descriptive findings on the dimensions of risk preparedness as identified by PCA. The different types of preparedness behaviour are distinctly distributed, but they have in common that the intention to prepare for risks in the future is clearly higher than the actually adopted behaviour.

In section 4.2, we show descriptive statistics of the most relevant influence factors on preparedness, which we identified by linear regression (section 4.3).

### 4.1 Dimensions of hazard risk preparedness

The four dimensions of IRM identified using PCA are: information gathering, social exchange, situational behaviour, and construction measures (Appendix Table A1).

Prevention starts with information gathering, which takes least effort compared to the other options, and is widely spread. The most common way to get informed is to follow forecasts and warnings (47%, see Fig. 1) in general.





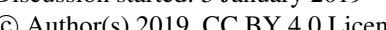

*Figure 1: Self-reported risk preparedness items: intention to adopt behaviour*



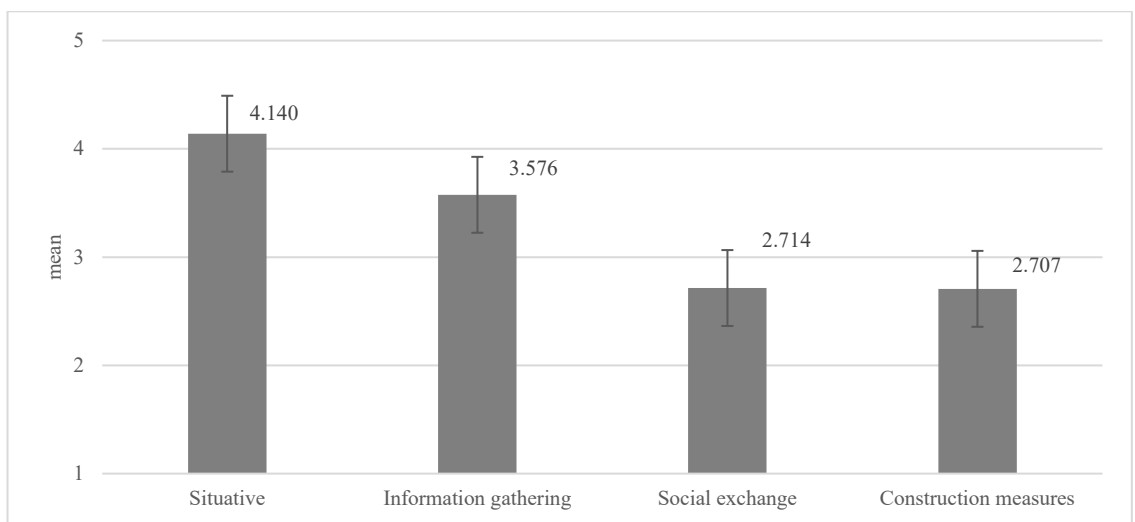

**Figure 2: Risk preparedness dimensions: elements that constitute risk preparedness, weighted by their importance (mean values)**

More effort taking behaviour than information gathering was measured by the dimension of social exchange, a distinctly communicative dimension. It includes talking to neighbours, which 7% (item 11) of the respondents already used to do. The option to join interactive events like participate in exercises is even less common. However, there appears to be a share of around 7% of the respondents who are actively engaged in natural hazard risk protection.

Most common of all kinds of behaviour (57%; item 9) is the intention to copy peer behaviour in case of an event. This points out a crucial influence of social surrounding on individual preparedness. Overall, these results show that there is preparedness and interest to get informed about natural hazards, but the respondents are less ready to take measures that take a lot of effort. Construction measures are the most expensive kind of protective behaviour in the list of preparedness items (Fig. 1 and 2). These questions applied to property owners only (n=1072). After all, 11% of the property owners had already implemented construction measures, and 4.6% reported to probably do so. While the intention to invest in temporary measures like removable installations is rather high (45%), few have done so. Strikingly low is the intention to consult an insurance: only 6.1% had done so already, almost every second owner (45%) reported intention to do so, but 48.6% refused such plans. These results indicate that the willingness to invest in prevention was rather low. Owners rather seem to rely on their insurance to sufficiently cover the risk (Fig 3).

**4.2 Reasons not to prepare**

Reasons not to prepare contribute to a better understanding of risk preparedness. Results are displayed in comparison of property owners and non-owners (Fig. 3). The main reason is that the respondents feel safe enough, and accept risk as a part of life. Another important reason was negative cost-benefit evaluation.

Figure 3 also shows the respondents' attitude towards safety. For instance, they rather bear damage than invest in safety. They also tend to think that they are in control of hazard risk, and trust the state to sufficiently ensure protection. Self-efficacy in



terms of feeling in control is wide spread: a majority of 63% assume that they can personally influence the impact caused by natural hazards.

These results indicate that the respondents generally give little priority to natural hazard protection. However, a vast majority of 77 describe themselves as generally cautious, which reveals a discrepancy in self-awareness and behaviour.

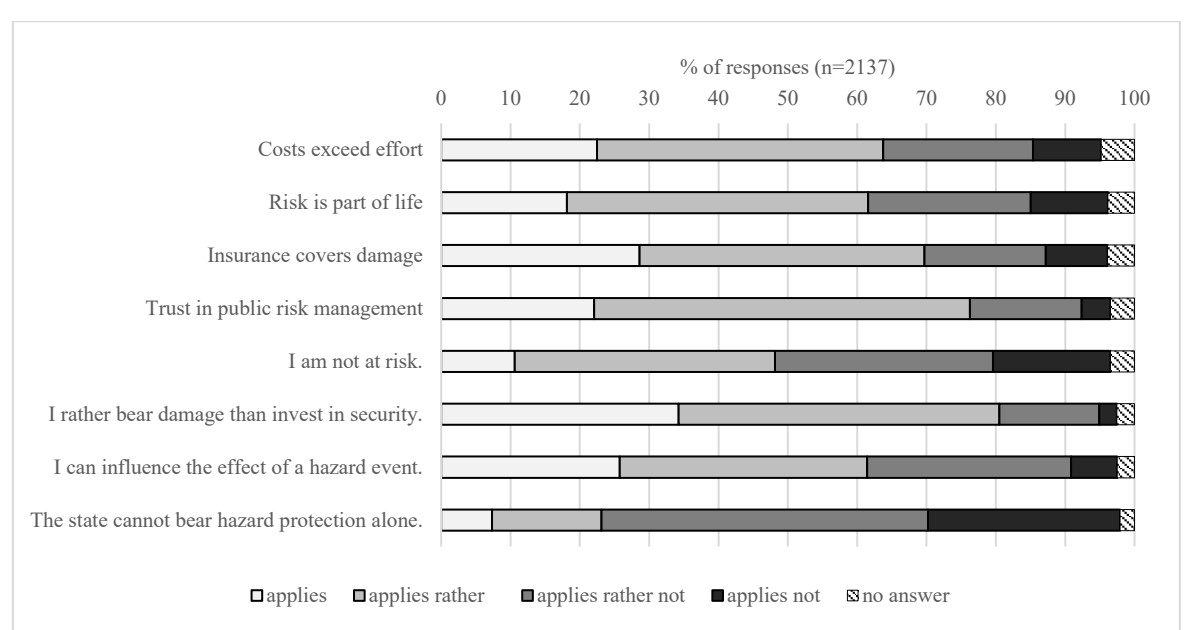

**Figure 3: Property owners' reasons not to implement measures, and their attitudes towards security**

## 4.3 High levels of awareness and trust

### 4.3.1 Risk awareness

Figure 4 shows the relative frequencies of respondents' answers to the 27 risk awareness items. Is also shows the five dimensions of risk awareness (see Appendix Table A1I), namely:

The 'concern/relevance' dimension (REL) comprises general interest in the topic of natural hazards, concern about hazards, and the assumption that damage by natural hazards will increase in the future. The items of the 'perceived probability' assessment dimension (PROB) could be combined to 'alpine hazards' and 'extreme weather events'. Floods and earthquakes,

however, are distinct in terms of perceived probability. The perceived probability of extreme weather events was assessed as rather high by almost half of all respondents. Although respondents show concern and awareness of the probability of hazard events, the feeling of personal threat is not frequent (around 10%). The dimension of personal threat (THRE) combines threat of personal damage of material loss and life danger. We differentiate whether respondents feel threatened by hazards while being at home, at work, during leisure time, or on the road. For instance, a share of 10% of the respondents feels threatened at

home 8.1%, and a considerably high share of 36 perceives high, resp. rather high threat during leisure activities.



**Figure 4: Risk awareness items: concern/relevance, perceived probability (PP) of natural hazard (NH) events, perceived personal danger (abbr. NH=natural hazard)**

5   **4.3.2 Hazard experience**

A majority of 62 of the respondents were exposed to natural hazards at least once. The other respondents were confronted with natural hazards by media reports only (Fig. 5). Property owners reported experience with hazards more frequently than others ($r$=.580, $p$=.0001). Out of all hazard experiences, most occurred during leisure activities (72.2%). Altogether, 29% of the





respondents felt personally endangered at least once, and another 30 had suffered material damage. In case that the experience did not involve material damage, the primary reaction is fascination.

We also recorded, how often respondents had a particular experience. Exposure during leisure activities was experienced by 16.8% of the respondents once, and 11 several times. Material damage (21.8% once; 7.7% several times) was mainly experienced by property owners: 14 of experienced damage once, and another 5.5% several times.

Regarding the effects of hazard experience (Table A3), results show that respondents frequently reported increased awareness and preparedness after an event. Apart from this it is a noticeable result that natural hazards are perceived as fascination by about half of the respondents. As Table A9 in the appendix shows, emotional affection by media reports was negatively correlated to awareness and preparedness. The relations between kinds and qualities of experience tell that the more direct the personal experience is, the more probable is an increase of self-reported awareness and preparedness. In all cases of experience, the event induced talking about natural hazards.

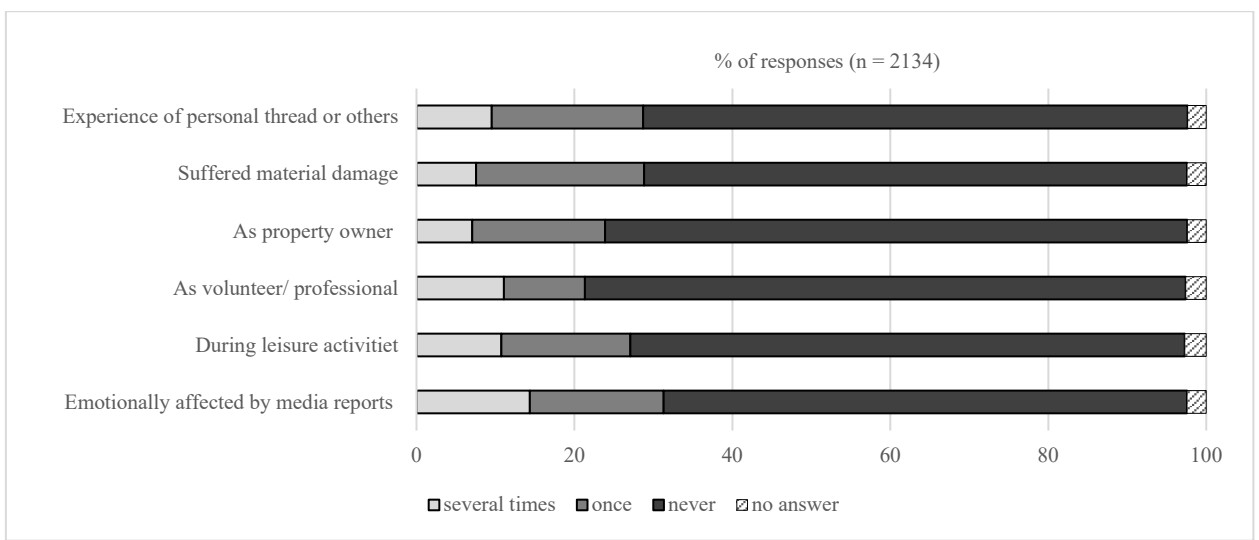

**Figure 5: Types of hazard experience**
Note: "as property owner" = having experienced a natural hazard event as owner of a property, but not necessarily as damage experience)

### 4.3.3 Trust, social integration, and responsibility

PCA showed that the items on trust in public risk management (A_IV) and social integration (A_V) measure one dimension each. We found a strikingly high level of trust in public risk management. Almost all respondents (92) believe that the authorities provide best possible protection from natural hazards. A majority of 68 reported that the authorities paid equal attention to different interests in hazard protection. Concerning social integration, most respondents reported positive results. A majority of 58 know many people in their community personally, and are active members in local associations. Civic engagement, however was less frequent: altogether only 23 often take the opportunity for participation.



For the perception of responsibility, three dimensions were found (A_VI): individual (self-responsibility, private actors), public actors (local, cantonal, federal), and emergency services (civil protection, fire brigade, police). Figure 6 shows that most respondents perceive responsibility as shared among different actors in natural hazard protection, since no actor is perceived not responsible. Authorities, however, are regarded as main responsible actors, followed by emergency services. Private
responsibility is assessed lower, but still high. The role of insurances is a distinct variable. They are regarded as the least responsible actors.

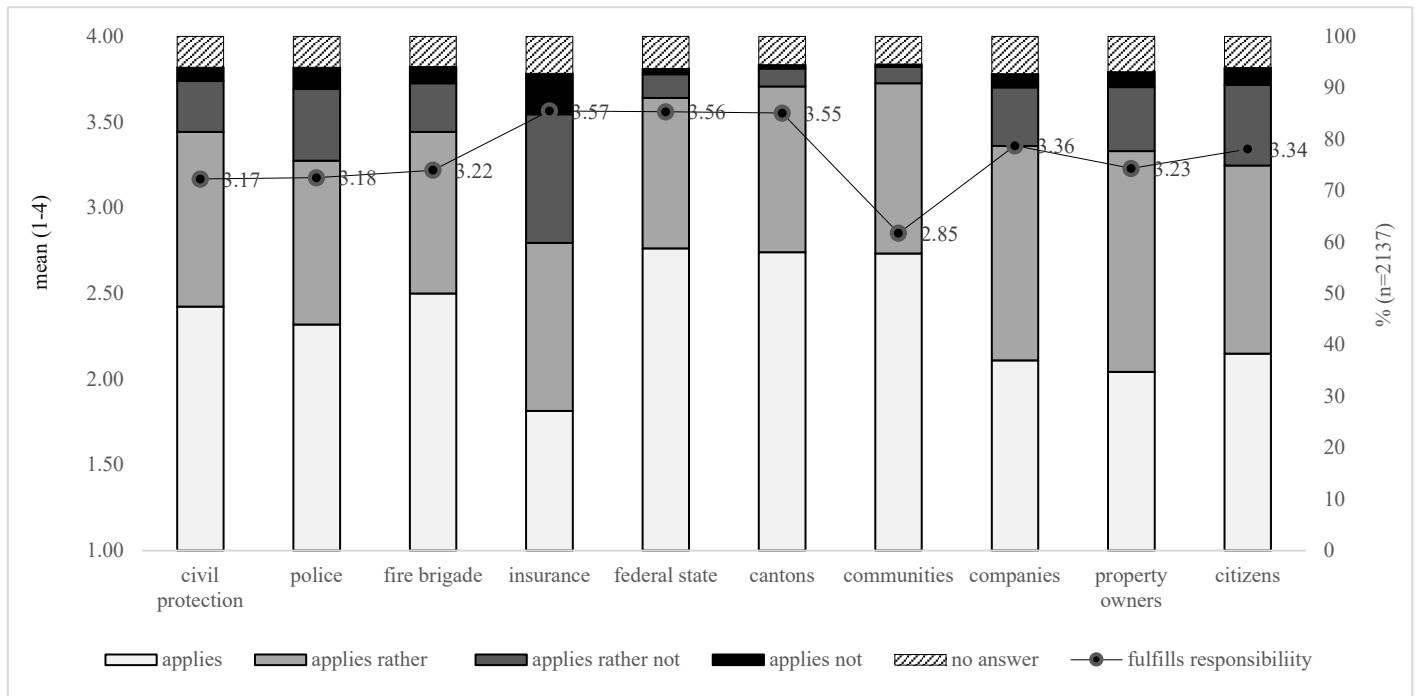

**Figure 6: Frequency of perceived responsibility of different actors (columns), and perceived fulfilment of this responsibility (line)**

Perceived responsibility was further analysed in relation to the other social capital variables. Results show that a high level of trust, resp. trust is significantly correlated to high levels of perceived responsibility of all actors except insurance. Private actors who perceive their own responsibility as high are well integrated in their communities. (Tab A_VII).

### 4.3.4 Influencing factors on risk preparedness

Linear regression analysis was conducted to investigate predictors of the four preparedness dimensions (Table 2). The main
predictors are: concern about natural hazards, the attitude that risk is part of life (negative influence), personal experience, perceived probability of severe weather events, and the attitude that the public institutions should prioritize safety over other values. Apart from these main influences that are significant for all four dimensions of preparedness, other explanatory variables were found that differently influence the preparedness dimensions.



For instance, social integration in the community, i.e., actively taking part in public community life and a personal network correlates with concern about neighbours.

Furthermore, attitudes and beliefs turned out to be influential: the belief in controllability, i.e. that the public authorities can control hazard risks, positively influences all individual preparedness dimensions, except situational behaviour: respondents with a belief in controllability show more intention to taking control of their individual risk as well. Another relevant attitude is perceived responsibility. Respondents who perceive responsibility for disaster risk reduction of other actors than themselves as high, show a higher level of individual preparedness. A third attitudinal influence is the New Ecological Paradigm (NEP; Dunlap, 2008; Anderson, 2012): a rather anthropocentric view goes along with a preference for implementing individual construction measures. Eco-centric values have a negative effect on active prevention. In this line of thought it appears better not to interfere in nature, because nature is in its complexity beyond human control and interference potentially counter-productive. The possibility of controlling damage is ascribed not only to authorities, science, or emergency services, but overall regarded as a common task. This is confirmed by the high correlation between perceived responsibilities of different actors.

Property ownership is the only social-economic predictor. Neither gender, nor the level of education or age directly turned out significant predictors of risk preparedness. Stronger predictors in the regression models suppress significant correlations in these respects. Nevertheless, property owners typically have a higher level of education, are older than non-owners, and more attached to their community than others. Ownership is also a wealth indicator that suppresses a direct influence of socio-economic status in the model.

Property ownership further significantly influenced active information behaviour. Owners rely less on copying other's behaviour in case of an emergency.

Surprisingly, perceived personal threat is only a minor predictor of all risk preparedness dimensions.

In the regression models, risk communication was not a significant influence on preparedness.



**Table 2: Linear regression models preparedness**

| Independent variable | Model | | | |
|---|---|---|---|---|
| | *Construction measures* | *Social exchange* | *Information behaviour* | *Situational behaviour* |
| "I am concerned about hazards." | .117*** | .152*** | .282*** | .116*** |
| "Risk is part of life." | -.169*** | -.123*** | -.083*** | -.062** |
| Sum of personal hazard experiences | .098*** | .089*** | .120*** | .088*** |
| Perceived probability of severe weather events | .085*** | .085*** | .083*** | .106*** |
| Attitude towards public measures: priority of safety | .09*** | .096*** | .106*** | .076** |
| Social integration | .150*** | .192*** | .161*** | n.s. |
| Controllability of damage by public measures | .106*** | .120*** | .047* | n.s. |
| "People in my community talk about natural hazards." | .104*** | .087*** | n.s. | -.049* |
| Perceived responsibility of insurance | .087*** | .129*** | n.s | -.075** |
| New ecological paradigm NEP | -.075*** | n.s | n.s | .074** |
| Property owner | .069*** | n.s | .055** | n.s. |
| Perceived self-responsibility | n.s | n.s | .066** | .082*** |
| Trust in public risk management | n.s | .051* | n.s | .05* |
| Perceived personal danger | .05* | .068** | n.s | n.s |
| | $R^2 = .254$, $F_{(14, 1836)} = 46.065$, $p <= .0001$ | $R^2 = .298$, $F_{(14, 1838)} = 57.052$, $p <= .0001$ | $R^2 = .259$, $F_{(14, 1839)} = 47.258$, $p <= .0001$ | $R^2 = .077$, $F_{(14, 1839)} = 12.000$, $p <= .0001$ |

*\*\*\* $p <= .001$; \*\* $p <= .01$; \* $p <= .05$*

## 5 Discussion

Building on the methodological development as laid out in section 3 above, the results of our nation-wide survey provide

5    insights on (1) influence factors on individual natural hazard risk preparedness among the common Swiss population (sect. 5.1), (2) support the hypothesis that no single dimension measures preparedness and (3) conclusions on capacity building towards a culture of risk (sect. 5.2).

As stressed in methods, regarding question 2, we emphasize the necessity of using a multi-dimensional approach to investigate risk preparedness. Accordingly, the operationalization of explanatory concepts like awareness and experience should not be

10   based on single items, or if so, then it needs to be clear, which dimension of a concept is measured. Results differ, if risk




preparedness is measured as an intention to prevent damage by hazards or actual behaviour. It further improves the comparability of survey results, if research designs regard different dimensions of preparedness like information gathering, situational behaviour, implementing construction measures, and social exchange. Similarly, risk awareness breaks down into relevance, perceived probability of events, and perceived threat.

## 5.1 Main predictors: experience, attitude, and integration

The strongest influences on hazard risk preparedness in the general population are individual attitudes towards risk, personal experience and community integration. The latter shows that in terms of natural hazard risk mitigation, the individual is to a relevant degree only as strong as the group. Risk awareness, however, turned out to play a less central role than assumed in previous literature (e.g. Bubeck et al., 2012; Bradford et al. 2012). Especially if it is one-dimensionally operationalized as perceived threat, there is little influence on individual risk preparedness. The influence of risk awareness is most significant as concern about hazards. According to our results, the influence of awareness is only important in the sense of concern. Further, results suggest that risk preparedness is so far not sufficiently understood due to a lack of systematic measurement.

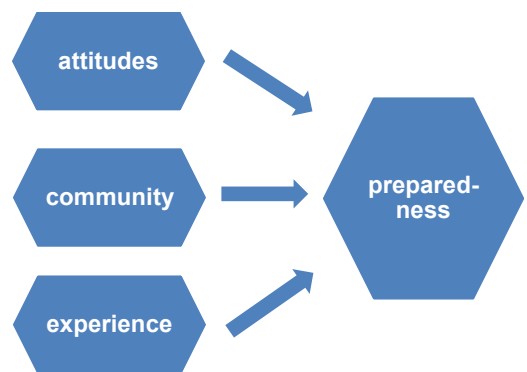

**Figure 7: Illustration of main influences on individual risk preparedness**

Attitudes are manifest in local communities. The broadly shared attitude that "Risk is part of life" is in contrast to the high level of self-reported risk aversion and reveals a cognitive dissonance. It is the main reported reason not to take any preventive action, but contradicts the high demand for security. We find such dissonance in research on risky health behaviour too (Freijy and Kothe, 2013). Denying a given severity of risk supports the justification that taking effort is not worth it.

The role of experience, however, is not as prominent as might be suggested. Future analysis should focus on a better understanding of how experience is processed and differentiate between types of experience. Based on our findings, we assume that the meaning ascribed to a particular experience is more important than its mere occurrence. Such meaning is apparently shaped in local talks, which itself turned out to be a predictor of IRM. Kasperson et al. (1988) already made clear that objective definitions of risk usually neglect social, psychological and cultural aspects. These include local cultures and shared attitudes



manifested in routines in interaction. Social integration, the third significant influence identified by the regression analysis is a most interesting resource in risk management and communication. Effective risk communication inspires people to talk about the subject in the first place. Communication needs to become dialogic to have a measurable effect. No variable of one-way communication turned out significant in the regression models. With Kuhlicke et al. (2011), we concluded that the combination

between instrumental and participatory approaches is recommendable.

## 5.2 Social capacity building

The results display a rather low level of IRP. However, it is important to differentiate between different types of preparedness, and the particular potential that lies in each in terms of capacity building. We see social integration at the core of effective and sustainable capacity building. As shown in this section, it enhances other capacities, too.

### 5.2.1 Improve situational behaviour options by information

Most common in all regions is situational behaviour, i.e., deciding how to behave in case of an event. Taking reasonable situational decisions depends on knowledge and experience. This can be achieved if people have relevant information at hand, and well internalized routines as described in existing capacity building literature (Kuhlicke et al, 2011). Our results newly show a tendency that people also orientate themselves towards other's behaviour. Therefore, we recommend providing

examples, both verbally (e.g. guidelines) and eventually by training motivated citizens. Considering to the Elaboration Likelihood Model (ELM) by Petty and Cacioppo (1986), we see that risk management requires an understanding of how information is processed as mentioned in the previous section.

In terms of information gathering, respondents use various means of available information. Spreading information diversely is more recommendable than relying on written forms of risk communication only. For positive information behaviour, concern

is a good predictor, and according to our results, concern itself is higher, if people in a community are used to talk about natural hazards. Therefore, animating such talks is a key capacity building measure.

### 5.2.2 Non-preventive behaviour and trust

Referring to Protection Motivation Theory PMT (Rogers 1975; 1997), we found that frustration resp. resignation are relevant non-protective responses to natural hazards. According to the high level of situational behaviour, immediate non-preventive

responses need more to be taken into consideration. Referring to Terpstra et al. (2011), who described a potentially hindering effect of trust on citizens' preventive behaviour, we explain passive behaviour as an effect of expectations towards the highly trusted public risk management. This is especially important, since a majority of the population has no individual damage experience. Feeling not concerned in combination with the feeling to be well protected is an obstacle for individual preparedness. Trust goes along with the feeling of being well protected, which reduces chances to take action (Visschers and

Siegrist, 2011; Terpstra and Gutterling, 2008). Therefore, practitioners should stress the importance of shared responsibility according to the integrated risk management paradigm.



Trust, further, has a twofold effect. It is a valuable resource in risk management as it enhances the chance that citizens take official information and warnings serious (Wachinger et al., 2013; Maidl and Buchecker, 2015). Thereby, fostering trust is a means of providing conditions for effective risk communication. At the same time, it is important to treat citizens as self-responsible actors, and for instance acknowledge that they set their priorities in a reasonable way to avoid paternalism (Demeritt, 2014). We suggest that critical thinking can give a counter-balance for the passive effect of trust. It can even raise chances for reasonable risk behaviour (Nakagawa, 2016).

### 5.2.3 Motivation to act and responsibility

The motivation to take preventive action is more difficult to achieve than getting attention to information. Generally, results show that the more effort a certain kind of behaviour takes, the lower the probability of its implementation. In case of construction measures, apart from attitudes, perceived responsibility of insurance plays an interesting role. Being supported by insurance raises the readiness to take preventive action. Perceived self-responsibility alone, however, has no significant influence on the readiness to implement construction measures. The most interesting result on responsibility is that it is considered to be shared. As theory suggests, of course individual risk preparedness requires a certain amount of self-responsibility (Beck, 1986). The process of individualization in risk society means 'responsibilization' and is a characteristic phenomenon in Western societies (Garland, 1996; Steinführer et al., 2008). Embedding our results into this finding supports the importance for all actors to engage in prevention, as according to the integrated risk management paradigm (PLANAT, 2004; Merz et al., 2010). Citizens who are aware of both the potential as well as the limitations of public and private risk management have better motivational pre-conditions to engage in protection and preparedness. Building a culture of risk therefore requires to strengthen a sense of self-efficacy (Kasperson et al., 1988; Renn, 2008; Breakwell, 2007).

### 5.2.4 Awareness raising beyond raising a feeling of threat

Regression results for all types of preparedness show that perceived threat is of minor relevance. This supports the risk paradox thesis that there is no strong direct link between awareness and preparedness (Wachinger et al., 2013). Natural hazard risks play a minor role in peoples' consciousness and daily lives. Results show that most respondents do not assume that hazard events, except of extreme weather events might occur in their neighbourhood. The relevance of hazard is perceived highest during leisure activities, and lowest, but still around 10% at home. This indicates a realistic judgement and a reasonable level of risk awareness. However, a low level of relevance means that information about natural hazards hardly can get people's attention. Instead, linking the topic of natural hazard protection with topics closer to daily live might increase response to information attempts. Information might be related to broader environmental topics, like human-nature relation, river restoration, recreation, or current topics in a community, especially land use planning. If interest in natural hazards is hardly given, symbolic information (Sütterlin and Siegrist, 2014) is a potential means of raising first interest. In case that the target population already is interested in natural hazards, an emphasis should be on providing supplementary facts that acknowledge citizens' own knowledge.



Neither perceived threat, nor high emotional affection by media reports significantly strengthen preparedness. This supports the conclusion that creating a feeling of threat in capacity building and risk communication is not of key importance. Additionally to ethical considerations on raising such emotions, it is important to further investigate, how addressees process information, e.g. under conditions of fear. Trumbo et al. (2007) found in the context of health risk research that high fear recipients are likely to believe and follow instructions. In social psychology this is called the 'central route to persuasion'. In contrast, under conditions of low fear and low probability of a hazard event, individuals rather develop a 'peripheral route' and a heuristic way to process information, e.g., credibility, or trust in the information sender (Petty and Capiocco, 1986).

### 5.2.5 Social integration and IRM

It is most important to consider the clear influence of social integration and local exchange about hazard related topics (Cutter et al., 2003). Social capacity building is related to cohesion and integration (Putnam, 2000). Knowing this, capacity building does include, apart from engaging dialogue about natural hazards, the development of community integration and encouragement of civic engagement.

Involving stakeholders requires open dialogue. Prior research indicates that participation in natural hazard risk management increases risk awareness and acceptance of measures (Wachinger et al., 2013). Authorities in many countries however tend to avoid encouraging such discourse (Pearce, 2003). In this respect, the culture of direct democracy and citizen involvement provides a ground for further developing a dialogic culture of risk, which may differ according to regional types of political culture in a federalist context like Switzerland.

Based on our finding that social relations matter a lot in capacity building, we assume that power relations, resp. empowerment of actors needs to be regarded, too. As Geoffrey et al. (2016) point out, power relations tend rather to be neglected in research on hazard risks. They are relevant in risk communication and the implementation of integrated risk management. In both situations, the average citizen and experts need to collaborate in mutual exchange about their perspectives and interests. In this sense, any attempt along the lines of the deficit model of risk communication (Demeritt, 2014) would be counter-productive as it puts citizens in a passive role towards authorities instead of fostering pro-active behaviour. Brown and Olofsson's approach (2014) of intersectional risk theory pays special attention to the recognition of such power relations. Their performative aspect in the social construction of risk is expressed by the term 'doing risk'. It highlights the everyday day cultural process of creating and dealing with risk. New approaches in risk management have the potential to renew inherited ways of risk construction not suitable to deal with challenges in the future. 'Doing risk' further better expresses the empowerment aspect of social capacity building.

Overall, social capacity building as theoretical framework allows to regard the interplay of motivational, knowledge, networks, institutional, economical, and procedural capacities (Kuhlicke et al., 2011; Aven and Renn, 2010). This forms the frame of culture of risk, which might vary among regions and municipalities, and requires adaptive capacity and holistic governance approaches (Aven and Renn, 2010; Gupta et al., 2010). In this, the individual knowledge and motivation to take preventive



action is embedded. In networks, common routines and attitudes are shared (Breakwell, 2007), and this highlights the importance of understanding the framing of local risk culture.

## 5.3 Limitations and recommendations for future research

A major limitation concerns representativeness. Despite using a representative sample of 10'000 household addresses, results cannot be considered representative, for 80% of the addressees did not respond to our survey. Results indicated, that there is a bias in people who are more than the average population interested in natural hazards, e.g. property owners.

For future research the use of multi-dimensional operationalization and measures of both risk awareness and preparedness is recommended. More consistent methods increase comparability of results and conclusions on cultural differences based on valid comparison. Our survey provides insights on items with valuable degree of information on risk preparedness, but we also found for instance that risk acceptance did not play a significant role in our analysis. Leaving out such items helps to reduce questionnaire length, which might increase response rates.

Measuring the effects of risk communication requires long-term measurement. This cross-sectional study provides insights into the effect of past communication on the present level of preparedness. Causal conclusions on the usage of the different communication means can only be drawn in a time series. In order to draw causal conclusions, it is important to repeat such surveys.

As mentioned in the previous section, little evidence is yet available on the effect of emotions in processing natural hazard risk information. Threat appraisal does not necessarily result in protective response, but occasionally cause cognitive dysfunctional non-protective behaviour like denial or frustration (Rogers 1975; 1997). Conversely, the feeling of self-efficacy and empowerment probably improves self-responsible pro-active behaviour. Generally, we suggest to empirically investigate the emergence and interplay of key preparedness factors individual attitudes, social integration, and experience.

We recommend to conduct research that identifies regional determinants of risk cultures in different contexts of political culture. This might also enhance understanding of generalizable determinants of a risk culture.

## 6 Conclusion

The findings of this nation-wide survey on individual risk preparedness in Switzerland for the first time provide empirical evidence on factors on which individual risk preparedness depends. Regarding methodology, we strongly recommend multi-dimensional measurement approaches that allow differentiation between dimension of key concepts like risk awareness and preparedness. This leads to clearer interpretation of results and enhances comparability of survey results.

We found that social exchange in local communities has a considerable influence on risk relevant attitudes. It is not only natural hazard experience that shapes the readiness to take preventive action, but also how such experience is processed and what conclusions are drawn with respect to future events. How people make sense of experience, as well as how they make sense of information about natural hazard is a matter of general attitudes towards risk. We also found dissonant attitudes, especially





high risk aversion in contrast to low readiness to take effort for protection, which we consider relevant to regard in strategies for capacity building. Integration and active participation in community life turned out to provide a fruitful ground to enhance preparedness. For instance, integration makes it more likely to gain important information through local communication chains. Accordingly, social exclusion increases the risk of suffering damage by natural hazard events.

5  Motivating citizens is as important as motivating other actors for the implementation of the integrated risk management (IRM) paradigm. Citizens regard responsibility as shared, and are motivated to engage if they see that other actors also fulfil their responsibility. Developing a sense of self-responsibility lies at the core of improving risk preparedness. Persons who generally show proactive behaviour are also more likely to take preventive actions with respect to natural hazards lead by example of others. Our results also showed that copying peer behaviour is a relevant response in emergency situations.

10 In terms of risk communication, results show that passively consumed (one-way) information is not a significant predictor of individual preparedness. Dialogue matters. As soon as natural hazard risks are talked about, information and opinions multiply especially on small scale community level and provide a solid basis for dealing with risk pre-emptively.



## Appendix

**Table A1: Principal Component analysis: risk preparedness dimensions**

| | Component | | | | |
|---|---|---|---|---|---|
| | Construction measures | Social exchange | Information gathering | Situational behaviour | Passivity (single item) |
| Regularly follow forecasts and warning | .042 | -.088 | .795 | -.035 | .114 |
| Do nothing until I get warned | .019 | .027 | .152 | -.015 | .965 |
| In case of storm stay away from trees | -.054 | -.133 | .502 | .460 | .199 |
| Participate in exercise/training | . 071 | .461 | .310 | -.132 | .062 |
| Get informed about alarm signals | -.060 | .134 | .755 | -.037 | .087 |
| Clarify how to behave in case of emergency | -.048 | .442 | .561 | -.110 | -.077 |
| Speak to neighbours/acquaintances | -.017 | .817 | .048 | .035 | -.041 |
| Clarify how quickly I can leave site | .041 | .634 | .184 | -.033 | -.148 |
| Behave as others do around me | -.069 | .833 | -.100 | .195 | .110 |
| Consult an insurance company | .390 | .423 | -.092 | .069 | .189 |
| Study risk map or the like | .070 | .265 | .481 | .022 | -.134 |
| Invest in temporary measures | .643 | .109 | -.024 | .071 | .000 |
| Move values to safe place in case of early warning | .423 | .002 | .135 | .243 | -.122 |
| In critical situation avoid risky sport | .023 | -.154 | .359 | .617 | -.181 |
| In critical situation first leave site. not rescue values | -.024 | .279 | -.279 | .804 | .023 |
| Professional consultation on possible damages | .861 | -.136 | .125 | -.048 | .036 |
| Construction measures | .950 | -.101 | -.044 | -.075 | -.010 |
| Work put emergency plan | .657 | .264 | -.111 | -.030 | -.013 |

Extraction Method: Principal Component Analysis. Rotation Method: Promax with Kaiser Normalization.



**Table A2: Principal component analysis: risk awareness dimensions**

| | Component | | | | |
|---|---|---|---|---|---|
| | Perceived probability of alpine hazard events | Concern/ relevance | Perceived personal danger (situations) | Perceived personal danger (damage) | Perceived probability of extreme weather events |
| I am generally interested in NH | .221 | .714 | .137 | .212 | .223 |
| People in my community talk a lot about NH. | .535 | .498 | .116 | .204 | -.076 |
| The impact of NH is often underestimated. | .145 | .479 | .175 | .153 | .157 |
| I am well informed about NH. | .115 | .627 | .265 | .024 | .136 |
| I am concerned about NH. | .215 | .750 | .165 | .294 | .037 |
| In Switzerland, damage by NH will increase in the future. | .127 | .723 | .178 | .129 | .17 |
| Perceived probability of floods | .528 | .251 | .149 | .172 | .217 |
| Perceived probability of landslides | .803 | .222 | .231 | .238 | .131 |
| Perceived probability of earthquakes | .157 | .149 | .387 | .002 | .517 |
| Perceived probability of rock falls | .849 | .147 | .218 | .227 | .121 |
| Perceived probability of avalanches | .791 | .132 | .199 | .156 | .042 |
| Perceived probability of storms | .194 | .261 | .193 | .366 | .721 |
| Perceived probability of heat waves/forest fires | .081 | .201 | .244 | .251 | .800 |
| Perceived probability of cold waves | .102 | .065 | .203 | .375 | .777 |
| Perceived danger of material damage | .229 | .188 | .345 | .822 | .335 |
| Perceived danger of personal damage (oneself or close person) | .261 | .223 | .410 | .796 | .309 |
| Perceived danger of economic damage. | .161 | .186 | .344 | .754 | .256 |
| Perceived danger at home | .283 | .283 | .687 | .478 | .237 |
| Perceived danger in spare time | .147 | .186 | .768 | .287 | .206 |
| Perceived danger in traffic/on the road | .235 | .197 | .760 | .284 | .203 |
| Perceived danger at work | .209 | .207 | .666 | .442 | .263 |

Extraction Method: Principal Component Analysis. Rotation Method: Promax with Kaiser Normalization.




**Table A3 Principal Component Analysis: quality of hazard experience**

|  | Effect on awareness | Effect on preparedness |
|---|---|---|
| I was fascinated. | .339 | -.266 |
| I was concerned. | .720 | .309 |
| I realised what impact NH can have. | .737 | .228 |
| I now think that damage caused by NH is self-induced. | .040 | .458 |
| I am more aware of risks caused by NH. | .715 | .481 |
| It motivated me to take preventive action. | .524 | .766 |
| It motivated me to get informed about NH. | .601 | .747 |
| Now I trust official warnings more than my own judgement. | -.303 | -.530 |
| I talked to others about the event. | .655 | .374 |

Extraction Method: Principal Component Analysis. Rotation Method: Promax with Kaiser Normalization

**Table A4 Principal Component Analysis: trust in public risk management**

| The authorities ... | Loading |
|---|---|
| …provide best possible protection of NH. | .783 |
| …take my interests serious. (n=2049) | .811 |
| …are competent in dealing with NH. | .829 |
| …work transparently. | .842 |
| …regard all interests equally. | .786 |

Extraction Method: Principal Component Analysis. Rotation Method: Promax with Kaiser Normalization

**Table A5 Principal Component Analysis: social integration**

|  | Loading |
|---|---|
| Active member of local associations | .770 |
| Regularly attend community meetings | .777 |
| Know a lot of people in my community | .783 |
| Often take opportunity for participation | .855 |

Extraction Method: Principal Component Analysis. Rotation Method: Promax with Kaiser Normalization



**Table A6 Principal Component Analysis: perceived responsibility**

|  | Loading | | |
|  | emergency services | state | self-responsibility |
|---|---|---|---|
| citizens | .047 | .097 | .839 |
| property owners | .106 | .149 | .893 |
| companies | .162 | .203 | .837 |
| communities | .178 | .864 | .245 |
| cantons | .176 | .929 | .152 |
| federal state | .201 | .896 | .099 |
| insurance | .566 | .227 | .174 |
| fire brigade | .921 | .097 | .063 |
| police | .904 | .120 | .086 |
| civil protection | .862 | .182 | .058 |

**Table A7: Bivariate correlations: trust in public risk management, social integration, and perceived responsibilities**

|  | insurance | trust in public risk management | social integration | self-responsibility (private actors) | responsibility of the state | responsibility of emergency services |
|---|---|---|---|---|---|---|
| insurance | 1 | .118** | n.s. | .240** | .331** | .467** |
| trust in public risk management |  | 1 | .084** | .102** | .144** | .186** |
| social integration |  |  | 1 | .112** | n.s. | n.s. |
| self-responsibility (private actors) |  |  |  | 1 | .354** | .228** |
| responsibility of the state |  |  |  |  | 1 | .349** |
| responsibility of emergency services |  |  |  |  |  | 1 |

5    ** p<.01





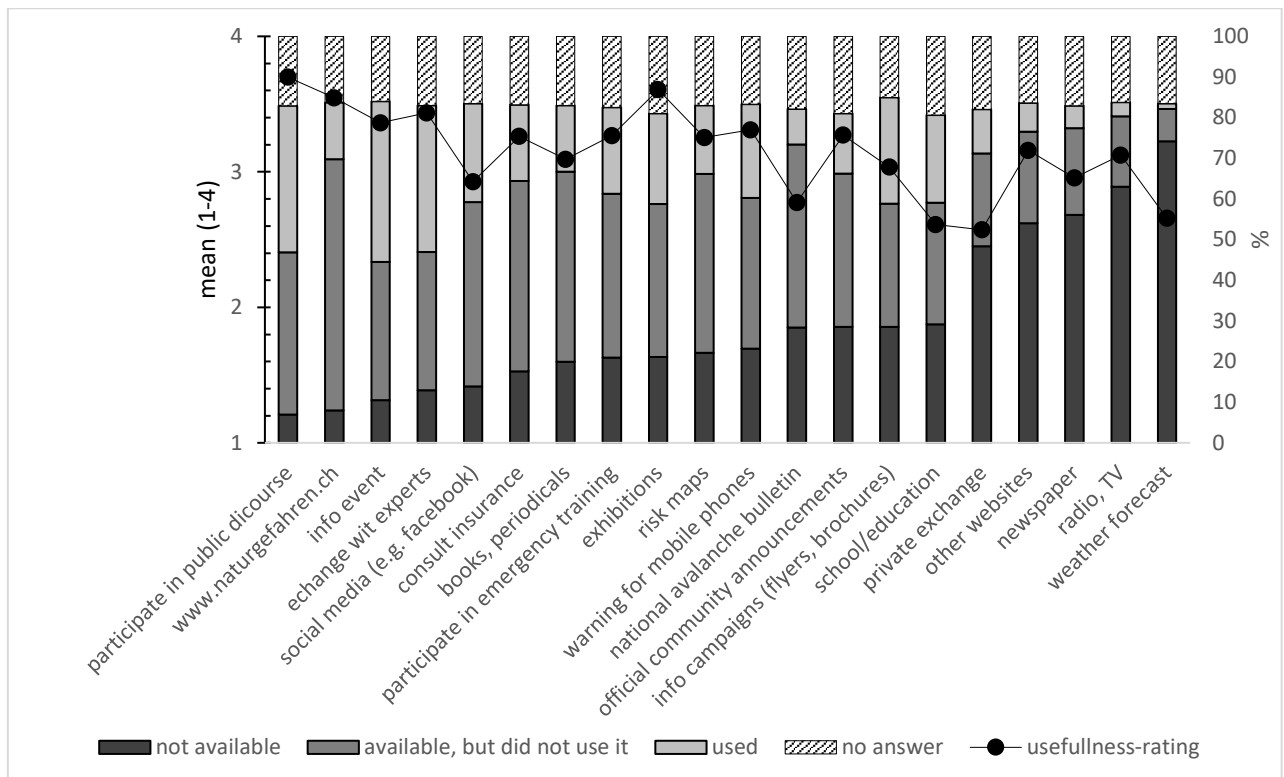

**Figure A1: Availability, usage, and rated usefullness of different communication means**

**Table A8: Crosstab: Experience of material damage and property ownership (n=2137)**

|       | never   | once    | several times | total    |
|-------|---------|---------|---------------|----------|
| yes   | 37.80%  | 7.90%   | 2.20%         | 47.90%   |
| no    | 32.70%  | 13.90%  | 5.50%         | 52.10%   |
| total | 70.50%  | 21.90%  | 7.70%         | 100.00%  |

**Table A8: Questionnaire overview: scales not used for analysis and single items**





**Table A9: Bivariate correlation: type and quality of respondents' natural hazard (NH) experience**

|  |  | Type of experience | | | | | |
|---|---|---|---|---|---|---|---|
|  |  | Emotionally affected by media reports | During leisure activity | As volunteer/ professional | As property owner | Suffered material damage | Experience of personal thread |
| Self-reported effect | I was fascinated | - | .162*** | .148*** |  | .065** | .106*** |
|  | I was concerned | -.340*** | .086*** | .061*** | .100*** | .103*** | .232*** |
|  | I realized what impact NH can have | -.306*** | .100*** | .065** | .094*** | .079*** | .136*** |
|  | I now think that damage caused by NH is self-induced | - | .051* | - | -.062** | - | - |
|  | I am more aware of risks caused by NH | -.300*** | .058** | - | .083*** | .062** | .088*** |
|  | It motivated me to take preventive action | -.207*** | .167*** | .193*** | .202*** | .218*** | .203*** |
|  | Now I trust official warnings more than my own judgement | -.244*** | .166*** | .173*** | .142*** | .143*** | .196*** |
|  | I talked to others about the event | .102*** | .051* | - | .081*** | .083*** | - |

*** p<.001; ** p<.01; *<.05; correlation coefficient: Spearman's rho.

## 5 Author contribution

Elisabeth Maidl as the main author wrote the original draft, Matthias Buchecker in the role of her supervisor, and David N. Bresch as external expert in natural hazard risk research reviewed and edited the draft.

In particular, Elisabeth Maidl and Matthias Buchecker mainly carried out conceptionalization, funding acquisition, methodology development, project administration, and provision of stud materials together. Elisabeth Maidl was in charge of data curation, statistical analysis, visualization and literature review.





### Acknowledgement

The authors thank their colleagues for continuing support and discussion, the cantonal and federal risk management practitioners, who participated in the questionnaire design workshop, the Federal Statistical Office for providing the household sample, and the referees for assisting in evaluating this paper. The study was financed in the frame of

the EU project KULTURisk.

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
