# Peer review of "Culture matters: Factors influencing natural hazard risk preparedness – a survey of Swiss households"

_Natural Hazards and Earth System Sciences, 2018_

## Referee Comment (RC1) · Anonymous Referee #1 · 29 Mar 2019

This paper on factors influencing natural hazard risk preparedness in Swiss households highlights the importance of multi-stakeholder communication for disaster preparedness at local levels. However, although the article prominently suggests that "culture matters" for disaster preparedness, it does not only lack any attempt to embed the term "culture" in the wider debate on risk preparedness but also fails to conceptualize the term as such. If the result of the survey is that "culture matters", how is it possible to not even mention "culture" in the conclusion? A potential way to address that fundamental conceptual problem of this article might be to simply avoid the use of the term "culture" as it has no analytical value added if it is not being defined or conceptualized.

---

## Short Comment (SC1) · 1 Apr 2019

Dear Reviewer #1

Thank you for commenting on our manuscript. I agree that the title and content of the manuscript should be consistent. I therefore will discuss replacing the term "culture" by "social integration" with my co-authors.

---

## Referee Comment (RC2) · Michael Nones (Referee) · 11 Sep 2019

The article presents the results of a nation-wide survey on factors influencing natural hazard risk preparedness, using Switzerland as a case study. The outcomes are interesting, but many drawbacks are affecting the manuscript and prevent its publication, as pointed out in the following.

One of the key points that reduces the validity of this manuscript is reported at the end of the manuscript (Section 5.3): the outcomes are statistically not significant, and there is more than a bias in the answers. Looking at the numbers, are you sure that around 20% of response rate (with only 15% in risk zones) and maximum R2 in the order of 0.25 could be considered as statistically significant? Moving from that, one

can conclude that all the results reported and their discussion are more a personal interpretation of the authors rather than scientific evidence. In addition, as also the authors highlighted, a single survey is not enough for drawing some conclusions on the general behaviour of a country. I suggest spending more effort in justifying what is the goal of your research, and why a reader should treat your outcomes as reliable and representative.

The manuscript is too vague, combining a multitude of concepts that are not well related and seem out of the context. In fact, in several points, the concept is described only under a general point of view, without adding any detail that can be helpful in better understanding the authors' view and the contribution that they want to provide in understanding such a concept. In this sense, a paper should not be a summary of the authors' knowledge, but rather a logical and consecutive description of specific research.

The rationale of the study and the theoretical description of the research (Section 2) should be rewritten, pointing out only the missing gaps addressed in this study, without mixing them with general theories. In addition, the use of very short subsections reduces the readability of the manuscript. Given that such sections are somehow interconnected, I suggest rearranging them for having a more coherent structure.

There is the need to be more specific and quantitatively address the topic. As an example, in asking about "suffered material damage" or "experience of personal threat" did you provide any indication? Each person can have a different reaction and consider threat or damage in different ways. How did you account for such inconsistencies among the citizens involved in the study?

The Discussion is just a continuation of the Results section, while it should propose new arguments that support (or not) the proposed approach. The conclusions that you reported are frequently not supported by the results (or a clear understanding of the connection between them is hindered behind complicated sentences).

In reporting numbers, it is not clear the units that you use. I think that, in most cases, you are speaking about percent, but, please, re-read carefully the text and provide all the details.

Before resubmitting, please double-check all the sentences, because, in this version, there are many phrases which are incomplete or not well connected with the successive ones.

The language style and grammar should be checked by a native English speaker.

———————————————